# HIERARCHICAL COMPOSITIONAL FEATURE LEARNING

**Miguel Lázaro-Gredilla, Yi Liu, D. Scott Phoenix, Dileep George**
Vicarious
San Francisco, CA, USA
{miguel,yiliu,scott,dileep}@vicarious.com

## ABSTRACT

We introduce the hierarchical compositional network (HCN), a directed generative model able to discover and disentangle, without supervision, the building blocks of a set of binary images. The building blocks are binary features defined hierarchically as a composition of some of the features in the layer immediately below, arranged in a particular manner. At a high level, HCN is similar to a sigmoid belief network with pooling. Inference and learning in HCN are very challenging and existing variational approximations do not work satisfactorily. A main contribution of this work is to show that both can be addressed using max-product message passing (MPMP) with a particular schedule (no EM required). Also, using MPMP as an inference engine for HCN makes new tasks simple: adding supervision information, classifying images, or performing inpainting all correspond to clamping some variables of the model to their known values and running MPMP on the rest. When used for classification, fast inference with HCN has exactly the same functional form as a convolutional neural network (CNN) with linear activations and binary weights. However, HCN's features are qualitatively very different.

## 1 INTRODUCTION

Deep neural networks coupled with the availability of vast amounts of data have proved very successful over the last few years at visual discrimination (Goodfellow et al., 2014; Kingma & Welling, 2013; LeCun et al., 1998; Mnih & Gregor, 2014). A basic desire of deep architectures is to discover the blocks –or features– that compose an image (or in general, a sensory input) at different levels of abstraction. Tasks that require some degree of image understanding can be performed more easily when using representations based on these building blocks.

It would make intuitive sense that if we were to train one of the above models (particularly, those that are generative, such as variational autoencoders or generative adversarial networks) on images containing, e.g. text, the learned features would be individual letters, since those are the building blocks of the provided images. In addition to matching our intuition, a model that realizes (from noisy raw pixels) that the building blocks of text are letters, and is able to extract a representation based on those, has found meaningful structure in the data, and can prove it by being able to efficiently compress text images.

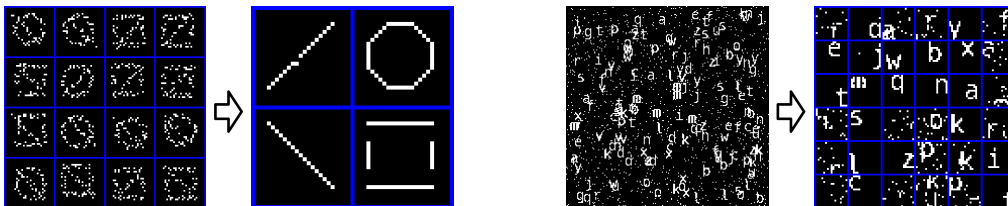

Figure 1: Features extracted by HCN. Left: from multiple images. Right: from a single image.

However, this is not the case with existing incarnations of the above models[1]. We can see in Fig. 1 the features recovered by the hierarchical compositional network (HCN) from a single image with no supervision. They appear to be reasonable building blocks and are easy to find for a human. Yet we are not aware of any model that can perform such apparently simple recovery with no supervision.

The HCN is a multilayer generative model with features defined at each layer. A feature (at a given position) is defined as the composition of features of the layer immediately below (by specifying their relative positions). To increase flexibility, the positions of the composing features can be perturbed slightly with respect to their default values (pooling). This results in a latent variable model, with some of the latent variables (the features) being shared for all images while others (the pool states) are specific for each image.

Comparing HCN with other generative models for images, we note that existing models tend to have at least one of the following limitations: a) priors are not rich enough; typically, the sources of variation are not distributed among the layers of the network, and instead the generative model is expressed as $X = f(Y) + \varepsilon$ where $Y$ and $\varepsilon$ are two set of random variables, $X$ is the generated image and $f(\cdot)$ is the network, i.e., the entire network behaves as a sophisticated deterministic function, b) the inference method (usually a separate recognition network) considers all the latent variables as independent and does not solve explaining away, which leads to c) the learned features being not directly interpretable as reusable parts of the learned images.

Although directed models enjoy important advantages such as the ability to represent causal semantics and easy sampling mechanics, it is known that the "explaining away" phenomenon makes inference difficult in these models (Hinton et al., 2006). For this reason, representation learning efforts have largely focused on undirected models (Salakhutdinov & Hinton, 2009), or have tried to avoid the problem of explaining away by using complementary priors (Hinton et al., 2006).

An important contribution of this work is to show that approximate inference using max-product message passing (MPMP) can learn features that are composable, interpretable and causally meaningful. It is also noteworthy that unlike previous works, we consider the weights (a.k.a. features) to be *latent variables* and not parameters. Thus, we do not use separate expectation-maximization (EM) stages. Instead, we perform feature learning and pool state inference jointly as part of the same message passing loop.

When augmented with supervision information, HCN can be used for classification, with inference and learning still being taken care of by a largely unmodified MPMP procedure. After training, discrimination can be achieved via a fast forward pass which turns out to have the same functional form as a convolutional neural network (CNN).

The rest of the paper is organized as follows: we describe the HCN model in Section 2; Section 3 describes learning and inference in the single layer and multilayer HCNs; Section 4 tests the HCN experimentally and we conclude with a brief discussion in Section 5.

## 2 THE HIERARCHICAL COMPOSITIONAL NETWORK

The HCN model is a discrete latent variable model that generates binary images by composing parts with different levels of abstraction. These parts are shared across all images. Training the model involves learning such parts from data as well as how to combine them to create each concrete image. The HCN model can be expressed as a factor graph consisting only of three types of factors: AND, OR and POOL. These perform the obvious binary operations and will be defined more precisely later in this section. The flexibility of the model allows training in supervised, semisupervised and unsupervised settings, including missing image data. Once trained, the HCN can be used for classification, missing value completion (pixel inference), sparsification, denoising, etc. See Fig. 2 for a factor graph of the complete model. Additional details of each layer type are given in Fig. 4.

At a high level, the HCN consists of a class layer at the top followed by alternating convolutional layers and pooling layers. Inside each layer there is a *sparsification*, a *representation* and *weights*

---

[1]Discriminative models find features that are good for classification, but not for generation (the training objective is not constrained enough). Existing generative models also fail at recovering the building blocks of an image because they either a) mix positive and negative weights (which turns out to be critical for them being trainable via backpropagation) or b) lack inference mechanisms able to perform explaining away.

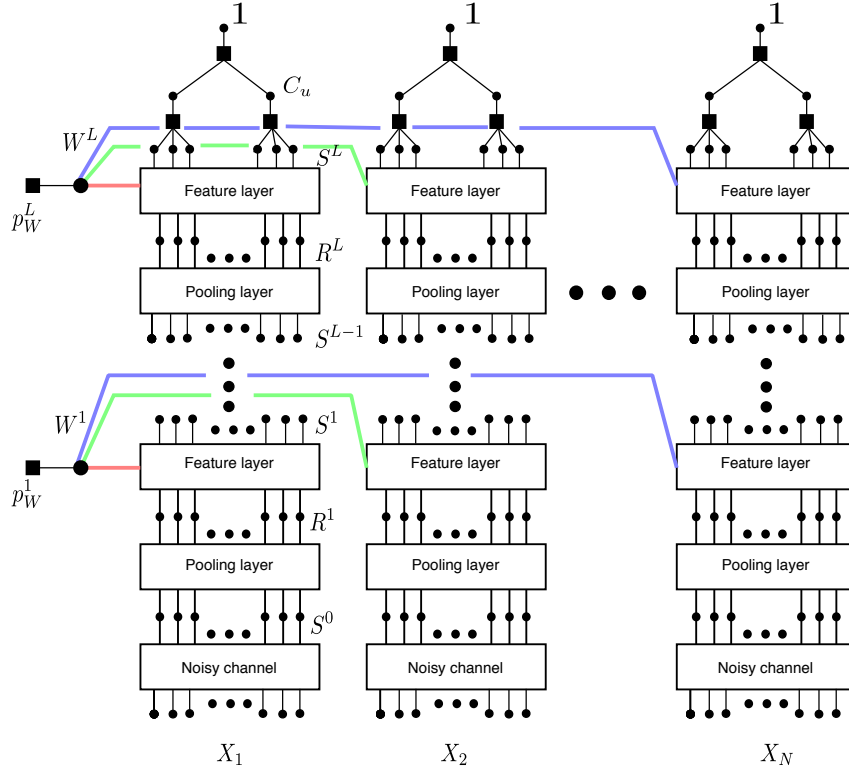

Figure 2: Factor graph of the HCN model when connected to multiple images $X_n$. The weights are the only variables that entangle multiple images. The top variables are clamped to 1 and the bottom variables are clamped to $X_n$. Additional details of each layer type are given in Fig. 4.

(a.k.a. features), each of which is a multidimensional array of latent variables. The class layer selects a category, and within it, which *template* is going to be used, producing the top-level sparsification. A sparsification is simply an encoding of the representation. A sparsification encodes a representation by specifying which features compose it and where they should be placed. The features are in turn stored in the form of *weights*. Convolutional layers deterministically combine the sparsification and the weights of a layer to create its representation. Pooling layers randomly perturb the position of the active elements (within a local neighborhood), introducing small variations in the process.

## 2.1 BINARY CONVOLUTIONAL FEATURE LAYER (SINGLE-LAYER HCN)

This layer can perform non-trivial feature learning on its own. We refer to it as a single-layer HCN. See Section 4.1 for the corresponding experiments.

In this case, since there is no additional top-down structure, a binary image is created by placing features at random locations of an image. Wherever two features overlap, they are ORed, i.e., if a pixel of the binary image is activated due to two features, it is simply kept active. We will call $W$ to the features, $S$ to the sparsification of the image (locations at which features are placed in that image) and $X$ to the image. All of these variables are multidimensional binary arrays.

The values of each of the involved arrays for a concrete example with a single-channel image is given in Fig. 3 (to display $S$ we maximize over $f$). The corresponding diagram is shown in Fig. 4.

In practice, each image $X$ is possibly multichannel, so it will have size $F_X \times H_X \times W_X$, where the first dimension is the number of channels in the image and the other two are its height and width. $S$ has size $F_S \times H_S \times W_S$, where the first dimension is the number of features and the other two are its height and width. We refer to an entry of $S_n$ as $S_{frc}$. Setting an entry $S_{frc} = 1$ corresponds to placing feature $f$ at position $(r, c)$ in the final image $X$. The features themselves are stored in $W$, which has size $F_W^{\text{below}} \times F_W \times H_W \times W_W$, where $F_W = F_S$ and $F_W^{\text{below}} = F_X$. I.e., each feature is a

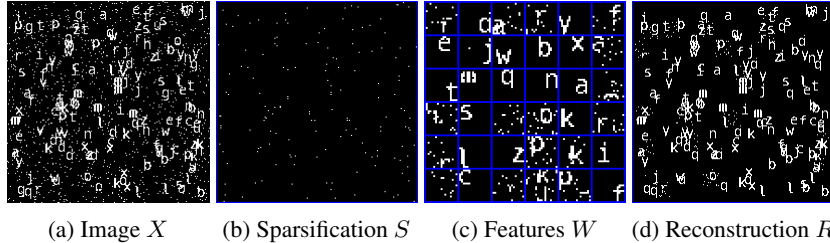

(a) Image $X$ (b) Sparsification $S$ (c) Features $W$ (d) Reconstruction $R$

Figure 3: Unsupervised analysis of image $X$ by a standalone convolutional feature layer of HCN.

small 3D array containing one of the building blocks of the image. Those are placed in the positions specified by $S$, and the same block can be used many times at different positions, hence calling this layer convolutional[2].

We can fully specify a probabilistic model for a binary images by adding independent priors over the entries of $S$ and $W$ and connecting those to $X$ through a binary convolution and a noisy channel. The complete model is

$$p(S) = \prod_{frc} p(S_{frc}) = \prod_{frc} p_S^{S_{frc}} (1 - p_S)^{1 - S_{frc}}$$

$$p(W) = \prod_{afrc} p(W_{afrc}) = \prod_{afrc} p_W^{W_{afrc}} (1 - p_W)^{1 - W_{afrc}} \tag{1}$$

$$p(X|R) = \prod_{arc} p_{\text{noisy}}(X_{arc}|R_{arc}) \text{ with } R = \text{bconv}(S, W) \text{ and } p_{\text{noisy}}(1|0) = p_{10}, p_{\text{noisy}}(0|1) = p_{01},$$

which depends on four scalar parameters $p_S, p_W, p_{01}, p_{10}$, controlling the density of features in the image, of pixels in each feature, and the noise of the channel, respectively. The indexes $a, f, r, c$ run over channels, features, rows and columns, respectively.

We have used the binary convolution operator $R = \text{bconv}(S, W)$. A binary convolution performs the same operation as a normal convolution, but operates on binary inputs and truncates outputs above 1. Our latent variables are arranged as three- and four-dimensional arrays, so we define $R = \text{bconv}(S, W)$ to mean $R_{a,:,:} = \min(1, \sum_f \text{conv2D}(S_{f,:,:}, W_{a,f,:,:}))$ where $\text{conv2D}(\cdot, \cdot)$ is the usual 2D convolution operator, $R$ and $S$ are binary 3D arrays and $W$ is a binary 4D arrays. The operator $\min(1, \cdot)$ truncates values above 1 to 1, performing the ORing of two overlapping features previously mentioned.

The binary convolution (and hence model (1)) can be expressed as a factor graph, as seen in Fig. 4. The AND factor can be written as $\text{AND}(b|t_1, t_2)$ and takes value 0 when the bottom variable $b$ is the logical AND of the two top variables $t_1$ and $t_2$. It takes value $-\infty$ in any other case. The OR factor, $\text{OR}(b|t_1 \ldots, t_M)$ takes value 0 when the bottom variable $b$ is the logical OR of the $M$ top variables $t_1 \ldots, t_M$. It takes value $-\infty$ in any other case.

When this layer is not used in standalone mode, but inside a multilayer HCN, the variables $R$ are connected to the pooling layer immediately below (instead of being connected to the image $X$ through the noisy channel) and the variables $S$ are connected to the pooling layer immediately above (instead of being connected to the prior).

## 2.2 THE CLASS LAYER

We assume for now that a single class is present in each image. We can then write

$$\log p(c_1, \ldots, c_K) = \text{POOL}(c_1, \ldots, c_K|1)$$

where $c_k$ are mutually exclusive binary variables representing which of the $K$ categories is present.

In general, we define $\text{POOL}(b_1, \ldots, b_M|t = 1) = -\log M$ when exactly one of the bottom variables $b_1, \ldots, b_m$ takes value 1 (we say that the pool is active), and $\text{POOL}(b_1, \ldots, b_M|t = 0) = 0$ when $b_m = 0 \; \forall_m$ (the pool is off). It takes value $-\infty$ in any other case.

---

[2]Additionally, the convolution implies the relations $H_X = H_W + H_S - 1$ and $W_X = W_W + W_S - 1$

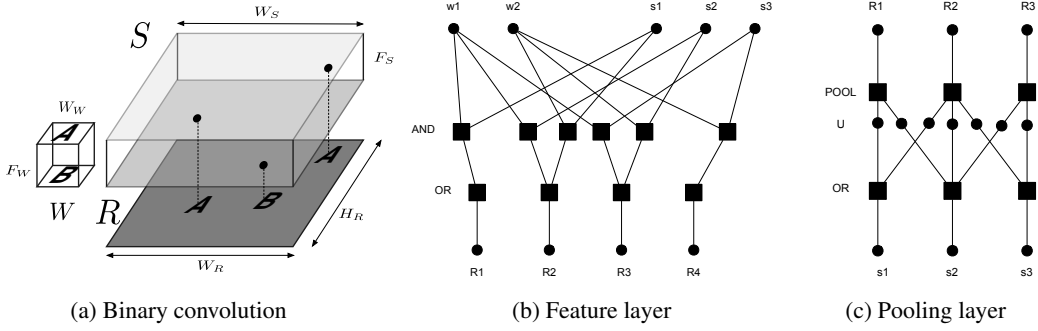

(a) Binary convolution (b) Feature layer (c) Pooling layer

Figure 4: Diagrams of binary convolution and factor graph connectivity for 1D image.

Within each category, we might have multiple templates. Each template corresponds to a different visual expression of the same conceptual category. For instance, if one category is furniture, we could have a template for chair and another template for table. Each category has binary variables representing each of the $J$ templates, $s_{jk}$ with $j \in [1 \dots J]$. If a category is active, exactly one of its templates will be active. The joint probability of the templates is then

$$\log p(S^L | c_1, \dots, c_K) = \sum_k \log p(s_{1k}, \dots, s_{Jk} | c_k) = \sum_k \text{POOL}(s_{1k}, \dots, s_{Jk} | c_k)$$

where these $JK$ variables are arranged as a 3D array of size $1 \times 1 \times JK$ called $S^L$ which forms the top-level sparsification of the template. A sample from $S^L$ will always have exactly one element set to 1 and the rest set to 0. Superscript $L$ is used to identify the layer to which a variable belongs. Since there are $L$ layers, $S^L$ is the top layer sparsification.

## 2.3 THE POOLING LAYER

In a multilayer HCN, feature layers and pooling layers appear in pairs. Inside layer $\ell$, the pooling layer $\ell$ is placed below the feature layer $\ell$.

Since the convolutional feature layer is deterministic, any variation in the generated image must come from the pooling layers (and the final noisy channel). Each pooling layer shifts the position of the active units in $R^\ell$ to produce the sparsification $S^{\ell-1}$ in the layer below. This shifting is local, constrained to a region of size[3] $\text{H}_P \times \text{W}_P \times 1$, the pooling window. When two or more active units in $R^\ell$ are shifted towards the same position in $S^{\ell-1}$, they result in a single activation, so the number of active units in $S^{\ell-1}$ is equal or smaller than the number of activations in $R^\ell$.

The above description should be enough to know how to sample $S^{\ell-1}$ from $R^\ell$, but to provide a rigorous probabilistic description, we need to introduce the intermediate binary variables $U_{\Delta r, \Delta c, f, r, c}$, which are associated to a shift $\Delta r, \Delta c$ of the element $R^\ell_{frc}$. The $\text{H}_P \text{W}_P$ intermediate variables associated to the same element $R^\ell_{frc}$ are noted as $U^\ell_{:,:,frc}$. Since an element can be shifted to a single position per realization and only when it is active, the elements in $U^\ell_{:,:,frc}$ are grouped into a pool

$$\log p(U^\ell | R^\ell) = \sum_{frc} \log p(U^\ell_{:,:,frc} | R^\ell_{frc}) = \sum_{frc} \text{POOL}(U^\ell_{:,:,frc} | R^\ell_{frc})$$

and then $S^{\ell-1}$ can be obtained deterministically from $U^\ell$ by ORing the $\text{H}_P \text{W}_P$ variables of $U$ that can potentially turn it on, $\log p(S^{\ell-1} | U^\ell) = \sum_{fr'c'} \log p(S^{\ell-1}_{fr'c'} | U^\ell) = \sum_{fr'c'} \text{OR}(S^{\ell-1}_{fr'c'} | \{U_{\Delta r, \Delta c, f, r, c}\}_{r':r+\Delta r, c':c+\Delta c})$. i.e., the above expression evaluates to 0 if the above OR relations are satisfied and to $-\infty$ if they are not.

---

[3]The described pooling window only allows for spatial perturbations, i.e., translational pooling. A more general pooling layer would also pool in the third dimension (Goodfellow et al., 2013), across features, which would introduce richer variation and also impose a meaningful order in the feature indices. Though we do not pursue that option in this work, we note that this type of pooling is required for a rich hierarchical visual model. In fact, the pooling over templates that we special-cased in the description of the class layer would fit as a particular case of this third-dimension pooling.

## 2.4 Joint probability with multiple images

The observed binary image $X$ corresponds to the bottommost sparsification[4] $S^0$ after it has traversed, element by element, a noisy channel with bit flip probabilities $p(X_{frc} = 1|S^0_{frc} = 0) = p_{10} < 0.5$ and $p(X_{frc} = 0|S^0_{frc} = 1) = p_{01} < 0.5$. This defines $p(X|S^0)$.

Finally, if we consider the weight variables to be independent Bernoulli variables with a fixed per-layer sparse prior $p^\ell_W$ that are drawn once and shared for the generation of all images, we can write the joint probability of multiple images, latent variables and weights as

$$\log p(\{X_n, H_n, C_n\}^N_{n=1}, \{W^\ell\}^L_{\ell=1}) = \sum_{\ell=1}^{L} \log p(W^\ell) + \sum_{n=1}^{N} \log p(X_n|S^0_n) + \log p(S^L_n|C_n) + \log p(C_n)$$

$$+ \sum_{n=1}^{N} \sum_{\ell=1}^{L} \log p(S^{\ell-1}_n|U^\ell_n) + \log p(U^\ell_n|R^\ell_n) + \log p(R^\ell_n|S^\ell_n, W^\ell)$$

where we have collected all the category variables $\{c_k\}$ of each image in $C_n$ and the remaining latent variables in $H_n$ and for convenience. Each image uses its own copy of the latent variables, but the weights are shared across all images, which is the only coupling between the latent variables.

The above expression shows how, in addition to factorizing over observations (conditionally on the weights), there is a factorization across layers. Furthermore, the previous description of each of these layers implies that the entire model can be further reduced to small factors of type AND, OR and POOL, involving only a few local variables each.

Since we are interested in a point estimate of the features, given the images $\{X_n\}^N_{n=1}$ and a (possibly empty)[5] subset of the labels $\{C_n\}^N_{n=1}$, we will attempt to recover the maximum a posteriori[6] (MAP) configuration over features, sparsifications, and unknown labels. Note that for classification, selecting $\{W^\ell\}^L_{\ell=1}$ by maximizing the joint probability is very different from selecting it by maximizing a discriminative loss of the type $\log p(\{C_n\}^N_{n=1}|\{X_n\}^N_{n=1}, \{W^\ell\}^L_{\ell=1})$, since in this case, all the prior information $p(X)$ about the structure of the images is lost. This results in more samples being required to achieve the same performance, and less invariance to new test data.

Once learning is complete, we can fix $\{W^\ell\}^L_{\ell=1}$, thus decoupling the model for every image, and use approximate MAP inference to classify new test images, or to complete them if they include missing data (while benefiting from the class label if it is available).

Even though we only consider the single-class-per-image setting, the compositional property of this model means that we can train it on single-class images and then, without retraining, change the class layer to make it generate (and therefore, recognize) combinations of classes in the same image.

## 3 Learning and inference

We will consider first the simpler case of a single-layer HCN, as described in Section 2.1. Then we will tackle inference in the multilayer HCN.

## 3.1 Learning in single-layer HCN

In this case, for model (1), we want to find

$$S^*, W^* = \arg\max_{S,W} p(X|S, W)p(S)p(W). \tag{2}$$

This is a challenging problem even in simple cases. In fact, it can be easily shown that boolean matrix factorization (BMF), a.k.a. boolean factor analysis, arises as a particular case of (2) in which the

---

[4]Alternatively, one could introduce the noisy channel between $R^0$ and $X$, but that would be equivalent to our formulation using a pooling window of size $1 \times 1 \times 1$ at the bottommost layer.

[5]The model was described as unsupervised, but the class is represented in latent variable $C_n$, which can be clamped to its observed value, if it is available.

[6]Note that we are performing MAP inference over discrete variables, where concerns about the arbitrariness of MAP estimators (see e.g., (Beal, 2003) Chapter 1.3) do not apply.

heights and widths of all the involved arrays are set to one. BMF is a decades-old problem proved to be NP-complete in (Stockmeyer, 1975) and with applications in machine learning, communications and combinatorial optimization. Another related problem is non-negative matrix factorization (NMF) (Lee & Seung, 1999), but NMF is additive instead of ORing the contributions of multiple features, which is not desired here.

One of the best-known heuristics to address BMF is the Asso (Miettinen et al., 2006). Unfortunately, it is not clear how to extend it to solve (2) because it relies on assumptions that no longer hold in the present case. The variational bound of (Jaakkola & Jordan, 1999) addresses inference in the presence of a noisy-OR gate and was successfully used in by (Šingliar & Hauskrecht, 2006) to obtain the noisy-OR component analysis (NOCA) algorithm. NOCA addresses a very similar problem to (2), the two differences being that a) the weight values are continuous between 0 and 1 (instead of binary) and b) there is no convolutional weight sharing among the features. NOCA can be modified to include the convolutional weight sharing, but it is not an entirely satisfactory solution to the feature learning problem as we will show. We observed that the obtained local maxima, even after significant tweaking of parameters and learning schedule, are poor for problems of small to moderate size.

We are not aware of other existing algorithms that can solve (2) for medium image sizes. The model (1) is directly amenable to mean-field inference without requiring the additional lower-bounding used in NOCA, but we experimented with several optimization strategies (both based in mean field updates and gradient-based) and the obtained local maxima were consistently worse than those of NOCA.

In (Ravanbakhsh et al., 2015) it is shown that max-product message passing (MPMP) produces state-of-the-art results for the BMF problem, improving even on the performance of the Asso heuristic. We also address problem (2) using MPMP. Even though MPMP is not guaranteed to converge, we found that with the right schedule, even with very slight or no damping, good solutions are found consistently.

Model (1) can be expressed both as a directed Bayesian network or as a factor graph using only AND and OR factors, each involving a small number of local binary variables. Finding features and sparsifications can be cast as MAP inference[7] in this factor graph.

MPMP is a local message passing technique to perform MAP inference in factor graphs. MPMP is exact on factor graphs without loops (trees). In loopy models, such as (1), it is an approximation with no convergence guarantees[8], although convergence can be often attained by using some damping $0 < \alpha \le 1$. See Appendix C for a quick review on MPMP and Appendix D for the message update equations required for the factors used in this work. Unlike Ravanbakhsh et al. (2015) which uses parallel updates and damping, we update each AND-OR factor[9] in turn, following a random in a sequential schedule. This results in faster convergence with less or no damping.

### 3.2 LEARNING IN MULTILAYER HCN (UNSUPERVISED, SEMISUPERVISED, SUPERVISED)

Despite its loopiness, we can also apply MPMP inference to the full, multilayer model and obtain good results. The learning procedure iterates forward and backward passes (a precise description can be found in Algorithm 1 below). In a forward pass, we proceed updating the bottom-up messages to variables, starting from the bottom of the hierarchy (closer to the image) and going up to the class layer. In a backward pass, we update the top-down messages visiting the variables in top-down order. Messages to the weight variables are updated only in the forward pass. We use damping only in the update of the bottom-up messages from a pooling layer during the forward pass. The AND-OR factors in the binary convolutional layer form trees, so we treat each of these trees as a single factor, since closed form message updates for them can be obtained. Those factors are updated once in random order inside each layer, i.e., sequentially. The pools at the class layer also from a tree, so we also treat them as a single factor. The message updates for AND, OR and POOL factors follow trivially from their definition and are provided in Appendix D.

---

[7]Note that we do not marginalize the latent variables (or the weights), but find their MAP configuration given a set of images. The sparse priors on the weights and the sparsification act as regularizers and prevent overfitting.

[8]MPMP works by iterating fixed point equations of the dual of the Bethe free energy in the zero-temperature limit. Convexified dual variants (see Appendix C) are guaranteed to converge, but much slower.

[9]Each OR factor is connected to several AND factors which together form a tree. We update the incoming and outgoing messages of the entire tree, since they can be computed exactly.

After enough iterations, weights are set to 1 if their max-marginal difference is positive and to 0 otherwise. This hard assignment converts some of the AND factors into a pass-through and the rest in disconnections. Thus the weight assignments define the connectivity between $S^\ell$ and $R^\ell$ on a new graph without ANDs. This is the learned model, that we can use to perform inference with with on new test images.

## 3.3 INFERENCE IN MULTILAYER HCN

Typical inference tasks are classification and missing value imputation. For classification, we find that *a single forward pass* seems good enough and further forward and backward passes are not needed (see Algorithm 1 for the description of the forward and backward passes). For missing value imputation a single forward and *top-down* pass is enough. In order to achieve higher quality explaining-away[10], we use a top-down pass instead of a backward pass. A top-down pass differs from a backward pass in that we replace step 5) with multiple alternating executions of steps 5) and 2). Therefore, it is not strictly a backward pass, but it proceeds top-down in the sense that once a layer has been fully processed, it is never visited again.

Interestingly, the functional form of the forward pass of an HCN is the same as that of a standard CNN, see Section 3.4, and therefore, an actual CNN can be used to perform a fast forward pass.

---

**Algorithm 1** Learning in Hierarchical Compositional Networks

---

**Input:** Hyperparameters $p_{01}, p_{10}, \{p_W^\ell\}_{\ell=1}^L$, data $\{X_n, C_n\}_{n=1}^N$ and network structure (pool and weight sizes for each layer)
**Init** Initialize bottom-up messages and messages to $\{W^\ell\}$ to zero. Initialize the top-down messages to $-\infty$. Initialize messages to $W$ from its prior uniformly at random in $(0.9 p_W, p_W)$ to break symmetry. Set constant bottom-up messages to $S^0$: $m(S_{frc}^0) = (k_1 - k_0)X_{frc} + k_0$ with $k_1 = \log \frac{1-p_{01}}{p_{10}}$ and $k_0 = \log \frac{p_{01}}{1-p_{10}}$
**repeat**
 Forward pass:
 **for** $\ell$ in $1, \ldots, L$ **do**
 1) Update messages from OR to $U^\ell$ in parallel
 2) Update messages from POOL to $R^\ell$ in parallel with damping $\alpha$
 3) Update messages from AND-OR to $W^\ell$ and $S^\ell$ sequentially in random order
 **end for**
 Update message from all class layer POOLs to $S^L$. Hard assign $C_n$ if label is available.
 Backward pass:
 **for** $\ell$ in $L, \ldots, 1$ **do**
 4) Update messages from AND-OR to $R^\ell$ sequentially in random order
 5) Update messages from POOL to $U^\ell$ in parallel
 6) Update messages from OR to $S^{\ell-1}$ in parallel
 **end for**
 Compute max-marginals by summing incoming messages to each variable
**until** Fixed point or iteration limit
**return** Max-marginal differences of $S^\ell$, $W^\ell$ and $R^\ell$

---

## 3.4 ABOUT THE HCN FORWARD PASS

### 3.4.1 FUNCTIONAL CORRESPONDENCE WITH CNN

After a single forward pass in an HCN (considering that the weights are known, after training), we get an estimate of the MAP assignment over categories. In practice, this assignment seems good enough for classification and further forward and backward passes are not needed.

The functional form of the first forward pass can be simplified because of the initial strongly negative top-down messages. Under these conditions, the message update rules applied to the pooling layers of the HCN have exactly the same functional form[11] as the max-pooling layer in a standard CNN. Similarly, applying the message update rule to the convolutional layers of the HCN —when the

---

[10] To avoid symmetry problems, instead of making the distribution of each POOL perfectly uniform, we can introduce slight random perturbations while keeping the highest probability value at the center of the pool. Doing so speeds up learning and favors centered backward pass reconstructions in the case of ties.

[11] See the Appendix D for the update rules of the messages of each type factor.

weights are known— has the same functional form as performing a standard (not binary) convolution of the bottom-up messages with the weights, just like in a standard CNN. At the top, the max-marginal over categories will select the one with the template with the largest bottom-up message. This can be realized with max-pooling over the feature dimension as done in (Goodfellow et al., 2013), or closely approximated using a fully connected layer and a softmax, as in more standard CNNs.

Simply put, the binary weights learned by an HCN can be copied to a standard CNN with linear activations and they will both produce the same classification results when we applied to the bottom-up messages (which are a positive scaling of the input data $X$ plus a constant).

### 3.4.2 Invariance to noise level

Consider we generate two data sets with the HCN model using the same weights but different bit-flip probabilities. If those probabilities are known, would we use different classifiers for each dataset? If we use a single forward pass, changing $p_{01}$ and $p_{10}$ produces a different monotonic transformation of all the bottom-up messages at every layer of the hierarchy, but the selected category, which *depends only on which variable has the largest value*, will not change. So, with a single-pass classifier, our class estimation does not change with the noise level. This has the important implication that an HCN does not need to be trained with noisy data to classify noisy data. It can be trained with clean data (where there is more signal and learning parts is easier) and used on noisy data without retraining.

## 4 Experiments

In the following, we experimentally characterize both the single-layer and multilayer HCN.

### 4.1 Single-layer HCN

We create several synthetic (both noisy and noiseless) images in which the building blocks –or features– are obvious to a human observer and check the ability of HCN to recover the them. The task is deceptively simple, and the existing the state of the art at this task, NOCA, is unable to solve several of our examples. Since the number of free parameters of the model is so small (3 in the case of a symmetric noisy channel), these can be easily explored using grid search and selected using maximum likelihood. The sensitivity of the results to these parameters is small.

HCN only requires straightforward MPMP with random order over the factors. For NOCA, initializing the variational posterior over the latent sources and choosing how to interleave the updates of this posterior with the update of the additional variational parameters (Šingliar & Hauskrecht, 2006) is tricky. For best results, during each E step we repeated the following 10 times: update the variational parameters for 20 iterations and then update the variational posterior (which is a single closed form update). The M update also required an inner loop of variational parameter updating.

The performance of HCN and NOCA can be assessed visually in Fig. 5. Column (a) shows each input image (these are single-image datasets) and the remaining columns show the features and reconstructions obtained by HCN and NOCA. In some of the input images we have added noise that flips pixels with 3% probability. For HCN (respectively NOCA), we binarize all the beliefs (respectively, variational posteriors) from the $[0, 1]$ range by thresholding at 0.5 and then perform a binary convolution to obtain the reconstruction. Because noise is not included in this reconstruction, a cleaner image may be obtained, resulting in unsupervised denoising (rows 1 and 4 of Fig. 5).

For a quantitative comparison, refer to Tab. 1. One algorithm-independent way to measure performance in the feature learning problem is to measure compression. It is known that to transmit a long sequence of $N$ bits which are 1 with probability $p$, we only need to transmit $NH(p)$ bits with an optimal encoding, where $H$ is the entropy. Thus sparse sequences compress well. In order to transmit these images without loss, we need to transmit either one sequence of bits (encoding the image itself) or three sequences of bits, one encoding the features, another encoding the sparsification and a last one encoding the errors between the reconstruction and the original image. Ideally, the second method is more efficient, because the features are only sent once and the sparsification and errors sequences are much sparser than the original image. The ratio between the two is shown together with running time on a *single* CPU. Unused features are discarded prior to computing compression.

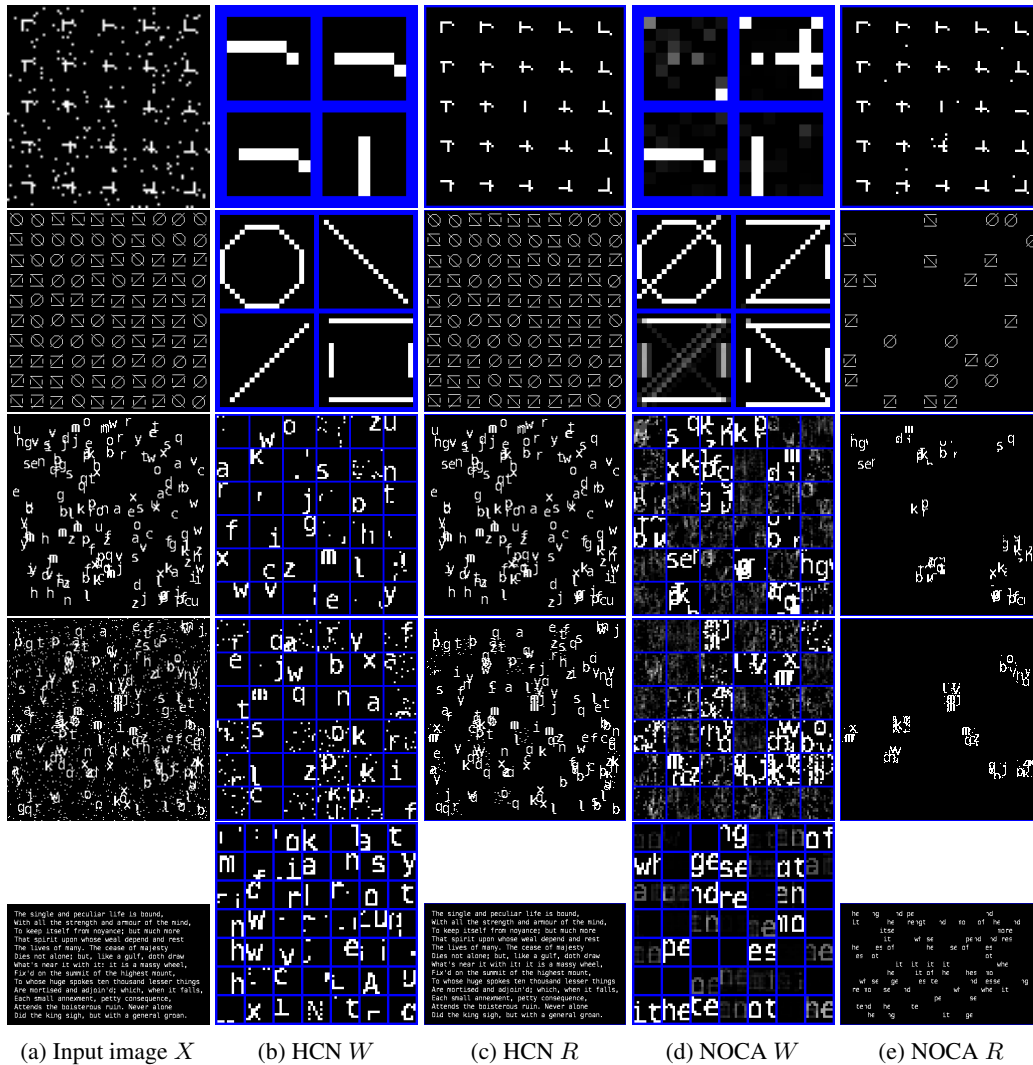

| (a) Input image $X$ | (b) HCN $W$ | (c) HCN $R$ | (d) NOCA $W$ | (e) NOCA $R$ |

Figure 5: Features extracted by HCN and NOCA and image reconstructions for several datasets. Best viewed on screen with zoom.

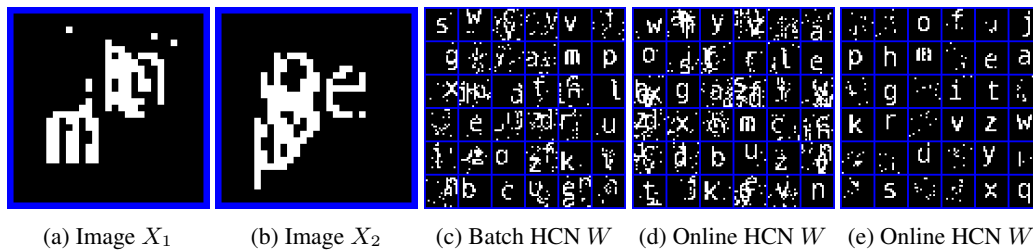

| (a) Image $X_1$ | (b) Image $X_2$ | (c) Batch HCN $W$ | (d) Online HCN $W$ | (e) Online HCN $W$ |

Figure 6: Online learning. (a) and (b) show two sample input images; (c) and (d) show the features learned by batch and online HCN using 30 input images and 100 epochs; (e) shows the features learned by online HCN using 3000 input images and 1 epoch.

| | Two bars | | Symbols | | Clean letters | | Noisy letters | | Text | |
|---|---|---|---|---|---|---|---|---|---|---|
| | comp. | time | comp. | time | comp. | time | comp. | time | comp. | time |
| NOCA | 84% | 0.67 m | 85% | 92 m | 98% | 662 m | 102% | 716 m | 84% | 1222 m |
| HCN | 83% | 0.07 m | 11% | 0.42 m | 38% | 25 m | 73% | 24 m | 28% | 31 m |

Table 1: Comp.: $E(X)/(E(S) + E(W) + E(X - R))$, where $E$ is the encoding cost. Time: minutes.

## 4.2 Online learning

The above experiments use a batch formulation, i.e., consider simultaneously all the available training data $\{X_n\}_1^N$. Since the amount of memory required to store the messages for MPMP scales linearly with the training data, this imposes a practical limit in the number of images that can be processed. In order to overcome this limit, we also consider a particular message update schedule in which the messages outgoing from factors connected to each image and sparsification $X_n, S_n$ are updated only once and therefore, after an image has been processed, can be discarded, since they are never reused. This effectively allows for online processing of images without memory scaling issues. Two modifications are needed in practice for this to work well: first, instead of processing only one image at a time, better results are obtained if the factors of multiple images (forming a minibatch) are processed in random order. Second, a forgetting mechanism must be introduced to avoid accumulating an unbounded amount of evidence from the processed minibatches.

In detail, the beliefs of the variables $W$ are initialized uniformly at random in the interval $(0.9 p_W, p_W)$ (we call these initial beliefs $b_{\text{prior}}^{(0)}(W_{afrc})$) and the beliefs of the variables $\{S_n\}_1^N$ are initialized to $p_S$. The initial outgoing messages from all the AND-OR factors are set to 0. Since each factor is only processed once, this allows implementing MPMP without ever having to store messages and only requiring to store beliefs. After processing the first minibatch using MPMP (with no damping), we call the resulting belief over each of the weights $b_{\text{post}}^{(0)}(W_{afrc})$ (as it standard for MPMP of binary variables, beliefs are represented using max-marginal differences in log space). Instead of processing the second minibatch using $b_{\text{post}}^{(0)}(W_{afrc})$ as the initial belief, we use $b_{\text{prior}}^{(1)}(W_{afrc}) = \lambda b_{\text{post}}^{(0)}(W_{afrc}) + (1 - \lambda) b_{\text{prior}}^{(0)}(W_{afrc})$, i.e., we "forget" part of the observed evidence, substituting it with the prior. This introduces an exponential weighing in the contribution of each minibatch. The forgetting factor is $\lambda \in (0, 1]$ specifies the amount of forgetting. When $\lambda = 1$ this reduces to normal MPMP (no forgetting), when $\lambda = 0$, we completely forget the previous minibatch and process the new one from scratch.

Fig. 6 illustrates online learning. HCN is shown 30 small images containing 5 randomly chosen and randomly placed characters with 3% flipping noise (see Fig. 5.(a) and (b) for two examples). They are learned in different manners. Fig. 5.(c): as a single batch with damping $\alpha = 0.8$ and using 100 epochs (each factor is updated 100 times); Fig. 6.(d): with minibatches of 5 images, no damping, $\lambda = 0.95$ and using 100 epochs; Fig. 6.(e): with minibatches of 5 images, no damping, $\lambda = 0.95$, using a single epoch, but using 3000 images, so that running time is the same.

## 4.3 Multi-layer HCN: synthetic data

We create a dataset by combining two traits: a) either a square (with four holes) or a circle and b) either a forward or a backward diagonal line. This results in four patterns, which we group in two categories, see Fig. 7.(a). Categories are chosen such that we cannot decide the label of an image based only on one of the traits. The position of the traits is jittered within a $3 \times 3$ window, and after combining them, the position of the individual pixels is also jittered by the same amount. Finally, each pixel is flipped with probability $10^{-3}$. This sampling procedure corresponds a 2-layer HCN sampling for some parameterization. We generate 100 training samples and 10000 test samples.

### 4.3.1 Unsupervised learning

We train the HCN as described in Section C on the 100 training data samples, not using any label information. We do set the architecture of the network to match the architecture generating the data. There are four hyperparameters in this model, $p_{01}, p_{10}, p_W^1, p_W^2$. Their selection is not critical. We

will choose them to match the generation process. MAP inference does discover and disentangle almost perfectly the compositional parts at the first and second layers of the hierarchy, see Figs. 7.(b) and 8.(a). In 8.(a), rows correspond with templates and columns correspond to each of the features of the first layer. We can see that the model has "understood" the data and can be used to generate more samples from it. Performing inference on this model is very challenging. We are not aware of any previous method that can learn the features of this simple dataset with so few samples. In other experiments we verified that, using local message passing as opposed to gradient descent was critical to successfully minimize our objective function. Results with the quality of Figs. 7.(b) and 8.(a) were obtained in every run of the algorithm. Running time is 7 min on a single CPU.

We can now clamp the discovered weights on both layers and use the fast forward pass to classify each training image as belonging to one of the four discovered templates (i.e., cluster them). We can even classify the test images as belonging to one of the four templates. When doing this, all the images in the training set get assigned to the right template and only 60 out of 10000 images in the test set do not get classified in the right cluster. This means that if we had just 4 labeled images, one from each cluster, we could perform 4-class minimally-supervised classification with just 0.6% error.

Finally, we run a single forward-backward pass of the inference algorithm on a test image with missing pixels. We show the inferred missing pixels in Fig. 7.(c). See also footnote 10.

### 4.3.2 SUPERVISED LEARNING

Now we retrain the model using label information. This results in the same weights being found, but this time the templates are properly grouped in two classes, as shown in Fig. 8.(a). Classification error on the test set is very low, 0.07%. We now compare the HCN classification performance with that of a CNN with the same functional form but trained discriminatively and with a standard CNN with ReLU activations, a densely connected layer and softmax activation. We minimize the crossentropy loss function with $L_2$ regularization on the weights. The test errors are respectively 0.5% and 2.5%, much larger than those of HCN. We then consider versions of our training set with different levels of pixel-flipping noise. The evolution of the test error is shown in Fig. 8.(c). For the competing methods we needed many random restarts to obtain good results. Their regularization parameter was chosen based on the test set performance.

### 4.4 MULTI-LAYER HCN: MNIST DATA

We turn now to a problem with real data, the MNIST database (LeCun et al., 1998), which contains 60000/10000 training/testing images of size $28 \times 28$. We want to generalize from very few samples, so we only use the first 40 digits of each category to train. We pre-process each image with a fixed set of 16 oriented filters, so that the inputs are a 16-channel image. We use a 2-layer HCN with 32 templates per class and 64 lower level features of size $26 \times 26$ and two layers of $3 \times 3$ pooling, $p_W^1 = 0.001, p_W^2 = 0.05$. These values are set a priori, not optimized. Then we test on both the regular MNIST training set and different corrupted versions[12] of it (same preprocessing

---

[12]See Appendix E for examples of each corruption type.

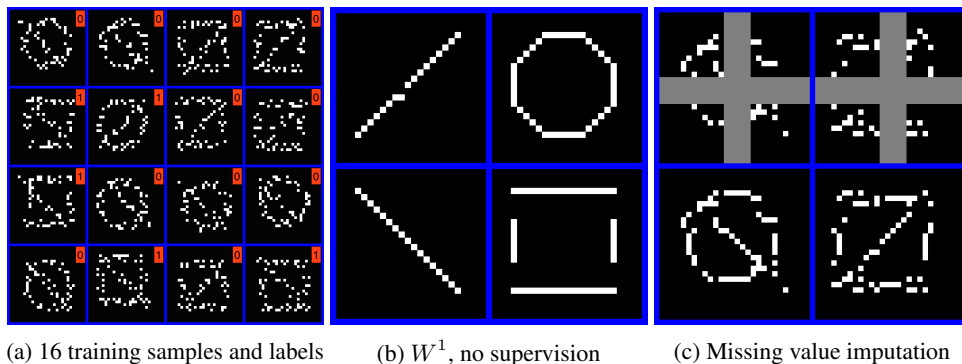

(a) 16 training samples and labels (b) $W^1$, no supervision (c) Missing value imputation

Figure 7: Samples from synthetic data and results from unsupervised learning tasks.

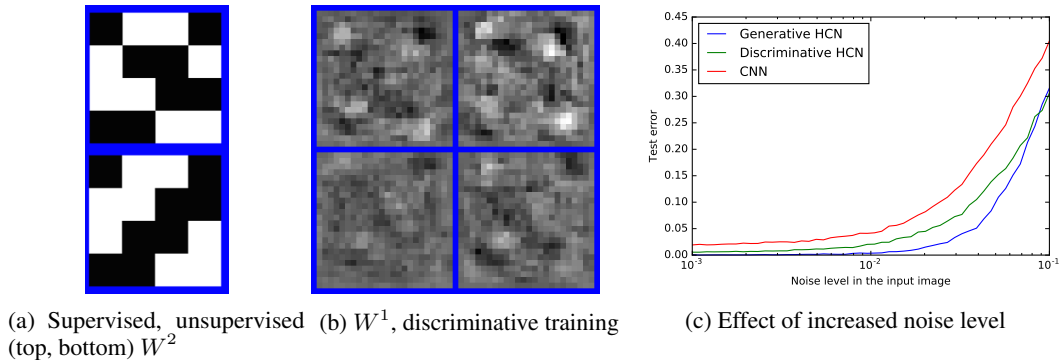

(a) Supervised, unsupervised (top, bottom) $W^2$

(b) $W^1$, discriminative training

(c) Effect of increased noise level

Figure 8: Discriminative vs. generative training and supervised vs. unsupervised generative training.

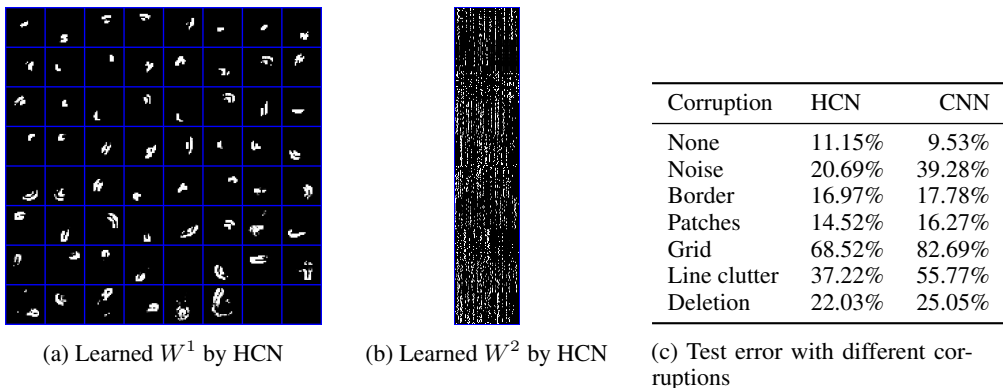

(a) Learned $W^1$ by HCN

(b) Learned $W^2$ by HCN

(c) Test error with different corruptions

| Corruption | HCN | CNN |
|---|---|---|
| None | 11.15% | 9.53% |
| Noise | 20.69% | 39.28% |
| Border | 16.97% | 17.78% |
| Patches | 14.52% | 16.27% |
| Grid | 68.52% | 82.69% |
| Line clutter | 37.22% | 55.77% |
| Deletion | 22.03% | 25.05% |

Figure 9: First layer of weights learned by HCN and CNN on the preprocessed MNIST dataset.

and no retraining). We follow the same preprocessing and procedure using a regular CNN with discriminative training and explore different regularizations, architectures and activation types, only fixing the pooling sizes and number of layers to match the HCN. We select the parameterization that minimizes the error on the clean test set. This CNN uses 96 low level features. Results for all test sets are reported on Fig. 9.(c). It can be seen that HCN generalizes better. The weights of the first layer of the HCN after training are shown in Fig. 9.(a). Notice how HCN is able to discover reusable parts of digits.

The training time of HCN scales exactly as that of a CNN. It is linear in each of its architectural parameters: Number of images, number of pixels per image, features at each layer, size of those features, etc. However, the forward and backward passes of an HCN are more complex and optimized code for them is not readily available as it is for a CNN, so a significant constant factor separates the running times of both. Training time for MNIST is around 17 hours on a single CPU. The RAM required to store all the messages for 400 training images in MNIST goes up to around 150GB. To scale to bigger training sets, an online extension (see Section 4.2) needs to be used.

# 5 CONCLUSIONS AND FUTURE WORK

We have described the HCN, a hierarchical feature model with a rich prior and provided a novel method to solve the challenging learning problem it poses. The model effectively learns convolutional features and is interpretable and flexible. The learned weights are binary, which is advantageous for storage and computation purposes (Courbariaux et al., 2015; Han et al., 2015). Future work entails adding more structure to the prior, leveraging more refined MAP inference techniques, exploring other update schedules and further exploiting the generalization-without-retraining capabilities of this model.

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

# A    RELATED WORK

There is a plethora of previous works that address hierarchical feature learning, usually in the setting or real-valued images, as opposed to binary ones: Fidler et al. (2014); Zhu et al. (2008; 2010); Wu et al. (2010); Si & Zhu (2013); Poon & Domingos (2011). Many of those works explicitly use AND-OR graphs, in the same spirit as our work. The most outstanding difference, however, between previous works and HCN is that HCN allows multiple features to overlap, thus creating new compositions. For instance, if feature H is a centered horizontal line and feature V is a centered vertical line, HCN can create a new feature "cross" that combines both, and the fact that both are overlapping and sharing a common active pixel (and many common inactive pixels) is properly handled. In contrast, previously cited models cannot overlap features, so they partition the input space and dedicate separate subtrees to each of them, and do so recursively. We can see in Figure 5, top row, how we can generate 25 different cross variations using only two features. This would not be possible with any of the cited models, which would need to span each combination as a separate feature. This fundamental difference makes HCN combinatorially more powerful, but also less tractable. Both learning and inference become harder because feature reuse introduces the well-known "explaining away" phenomenon (Hinton et al., 2006).

As a side, note the difference between the meaning of "OR" as used in the present work and in previous works on AND-OR graphs: what they call "OR", is what we term POOL (an exclusive bottom-up OR of elements), whereas HCN has a novel third type of gate, the "OR" connection (a non-exclusive, top-down OR of elements) to be able to handle explaining away. Standard AND-OR (or more clearly, AND-POOL) graphs lack the top-down ORing and therefore are not able to handle explaining away.

In the compositional hierarchies of Fidler et al. (2014), the lack of feature reuse allows for inference to be exact, since the graphical model is tree-like. Features are learned using a heuristic that relies on the exact inference, similar in spirit to EM. The AND-OR template learning methods of (Zhu et al., 2008; 2010) use respectively max-margin and incremental concave-convex procedures to optimize a discriminative score. Therefore they require supervision (unlike HCN) and a tractable inference procedure (to make the discriminative score easy to optimize), which again is achieved by not allowing overlapping features. The sum-product networks (SPNs) of (Poon & Domingos, 2011) express features as product nodes. In order to achieve feature overlapping, two product nodes spanning the same set of pixels (but with possibly different activation patterns) should be active simultaneously. This would violate the consistency requirement of SPNs, making HCN a more compact way to express feature overlap[13] (with the price to be paid being lack of exact inference).

The AND-OR template (AOT) learning of (Wu et al., 2010) again cannot deal properly with the generation of superimposed features, having to create new features to handle every combination. In Section B we will compare AOT feature learning and HCN feature learning and check how these limitations make AOT unable to disentangle the generative features.

Grammars exclude the sharing of sub-parts among multiple objects or object parts in a parse of the scene (Jin & Geman, 2006), and they limit interpretations to single trees even if those are dynamic (Williams & Adams, 1999). Our graphical model formulation makes the sharing of lower-level features explicit by using local conditional probability distributions for multi-parent interactions, and allows for MAP configurations (i.e, the parse graphs) that are not trees.

The deep rendering model (DRM) of Patel et al. (2015) is, to some extent, a continuous counterpart of the present work. Although DRMs allow for feature overlap, the semantics are different: in HCN the amount of activation of a given pixel is the same whether there are one or many features (causes) activating it, whereas in DRM the activation is proportional to the number of causes. This means that the difference between DRM and HCN is analogous to the difference between principal component analysis and binary matrix factorization: while the first can be solved analytically, the second is hard and not analytically tractable. This results in DRM being more tractable, but less appropriate to handle problems with binary events with multiple causes, such as the ones posed in this paper.

Two popular approaches to handle learning in generative models, largely independent of the model itself, are variational autoencoders (VAEs) and generative adversarial networks (GANs). We are not

---

[13]An exponentially big SPN could indeed encode an HCN.

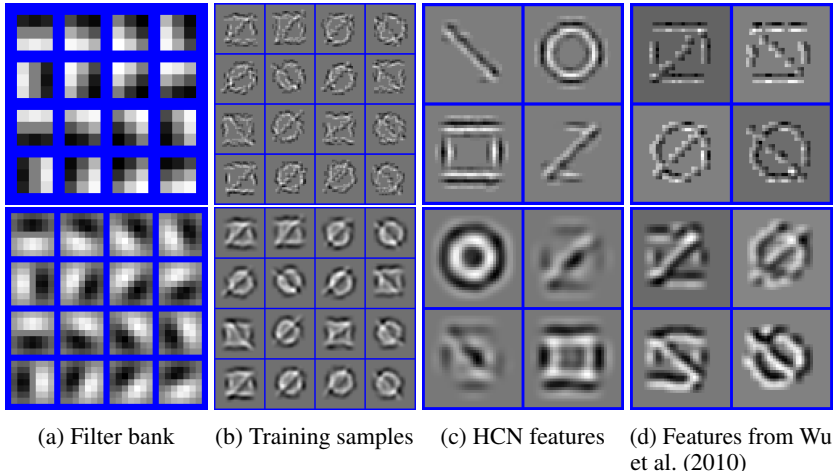

| (a) Filter bank | (b) Training samples | (c) HCN features | (d) Features from Wu et al. (2010) |

Figure 10: Results of training a modified HCN on a grayscale image. A filter bank is convolved with the input image to provide the bottom up messages to each channel of HCN. The filter bank sizes in this simple example are adapted to match those of generation. As a benchmark, Wu et al. (2010) is used on the same data and is also given knowledge of the filter bank in use. Top row: $3 \times 3$ filter size. Bottom row: $7 \times 7$ filter size.

aware of any work that uses a VAE or GAN with a generative model like HCN and such an option is unlikely to be straightforward.

Most common VAEs rely on the reparameterization trick for variance reduction. However, this trick cannot be applied to HCN due to the discrete nature of its variables, and alternative methods would suffer from high variance. Another limitation of VAEs wrt HCN is that they perform a single bottom-up pass and lack of explaining away: HCN combines top-down and bottom-up information in multiple passes, isolating the parent cause of a given activation, instead of activating every possible cause.

GANs need to compute $\nabla_W D(G_W(\varepsilon))$ where $D(\cdot)$ is the discriminative network and $G_W(\varepsilon)$ is a generative network parameterized by the features $W$. In this case, not only $W$ is binary, but also the generated reconstructions at every layer, so the GAN formulation cannot be applied to HCN as-is. One could in principle relax the binary assumption of features and reconstructions and use the GAN paradigm to train a neural network with sigmoidal activations, but it is unclear that the lack of binary variables will still produce proper disentangling (the convolutional extension of NOCA also has this problem due to the use of non-binary features and produces results that are inferior to HCN).

## B  COMBINING WITH GRAYSCALE PREPROCESSING

The HCN is a binary model. However, to process real-valued data, it can be coupled with an initial grayscale-to-binary preprocessing step to do feature detection. We tested this by generating a grayscale version of our toy data and then computing the bottom-up messages to $S^0$ by convolving the input image with a filter bank. This is roughly equivalent to replacing the noisy binary channel of HCN with a Gaussian channel. We used 16 preprocessing filters, which means that $S^0$ has 16 channels. 200 training images (unsupervised) were used. Two filter sizes, $3 \times 3$ and $7 \times 7$ were tested. We also run the AOT feature learning method of Wu et al. (2010) on the same data for comparison. The results of training on 200 training images (unsupervised) is provided in Figure 10. When the larger filter is used, the diagonal bars are harder to identify so their disentangling is poorer.

## C  MAX-PRODUCT MESSAGE PASSING (MPMP)

The HCN model can be expressed both as a directed Bayesian network or as a factor graph using only POOL, AND, and OR factors, each involving a small number of local binary variables. Both

learning and ulterior classification can be cast as MAP inference in this factor graph. Other tasks, such as filling in unknown image data can also be performed by MAP inference.

MAP inference can be performed exactly on factor graphs without loops (trees) in linear time, but it is an NP-hard problem for arbitrary graphs (Shimony, 1994). The factor graph describing our model is highly structured, but also very loopy.

There is large body of works (Wang & Daphne, 2013; Meltzer et al., 2009; Globerson & Jaakkola, 2008; Kolmogorov, 2006; Werner, 2007), addressing the problem of MAP inference in loopy factor graphs. Perhaps the simplest of these methods is the max-product algorithm, a variant of dynamic programming proposed in (Pearl, 1988) to find the MAP configuration in trees.

The max-product algorithm defines a set of *messages* $m_{a \to i}(y_i)$ going from each factor $a$ to each of its variables $y_i$. The sum of the messages incoming to a variable $\mu(y_i) = \sum_{a:y_i \in y_a} m_{a \to i}(y_i)$ defines its approximate max-marginal[14] $\mu(y_i)$. The max-product algorithm then proceeds by updating the outgoing messages from each factor in turn so as to make the approximate max-marginals consistent with that factor. This algorithm is not guaranteed to converge if there are loops in the graph, and if it does, it is not guaranteed to find the MAP configuration. Damping the updates of the factors has been shown to improve convergence in loopy belief propagation (Heskes, 2002) and was justified as local divergence minimization in (Minka et al., 2005). Using a damping factor $0 < \alpha \leq 1$ for max-product, the update rule is

$$m_{a \to i}^{t+1}(y_i) = (1 - \alpha)m_{a \to i}^t(y_i) + \alpha \max_{y_{a \setminus i}} \log \phi_a(y_i, y_{a \setminus i}) + \sum_{y_j \in y_{a \setminus i}} m_{a \to j}^t(y_j) + \kappa \qquad (3)$$

and the original update rule is recovered for $\alpha = 1$. The value $\kappa$ is arbitrary and does not affect the algorithm. We select it to make $m_{a \to i}^{t+1}(y_i = 0) = 0$, so that messages can be stored as a single scalar. When storing messages in this way, their sum provides the max-marginal difference, which is enough for our purposes.

Eq. (3) can be computed exactly for the three type of factors appearing in our graph, so message updating can be performed in closed form. Despite the graph of our model being very loopy, it turns out that a careful choice of message initialization, damping and parallel and sequential updates produces satisfactory results in our experiments. For further details about max-product inference and MAP inference via message passing in discrete graphical models we refer the reader to (Koller & Friedman, 2009).

## D  MAX-PRODUCT MESSAGE UPDATES FOR AND, OR AND POOL FACTORS

In the following we provide the message update equations for the different types of factors used in the main paper. The messages are in normalized form: each message is a single scalar and corresponds to the difference between the unnormalized message value evaluated at 1 and the unnormalized message value evaluated at 0. For each update we assume that the incoming messages $m_{IN}(\cdot)$ for all the variables of the factor are available. The incoming messages are the sum of all messages going to that variable except for the one from the factor under consideration.

The outgoing messages are well-defined even for $\pm\infty$ incoming messages, by taking the corresponding limit in the expressions below.

### D.1  AND FACTOR

Bottom-up messages

$$m_{OUT}(t_1) = \max(0, m_{IN}(t_2) + m_{IN}(b)) - \max(0, m_{IN}(t_2))$$
$$m_{OUT}(t_2) = \max(0, m_{IN}(t_1) + m_{IN}(b)) - \max(0, m_{IN}(t_1))$$

Top-down message

$$m_{OUT}(b) = \min(m_{IN}(t_1) + m_{IN}(t_2), m_{IN}(t_1), m_{IN}(t_2))$$

---

[14]The max-marginal of a variable in a factor graph gives the maximum value attainable in that factor graph for each value of that variable.

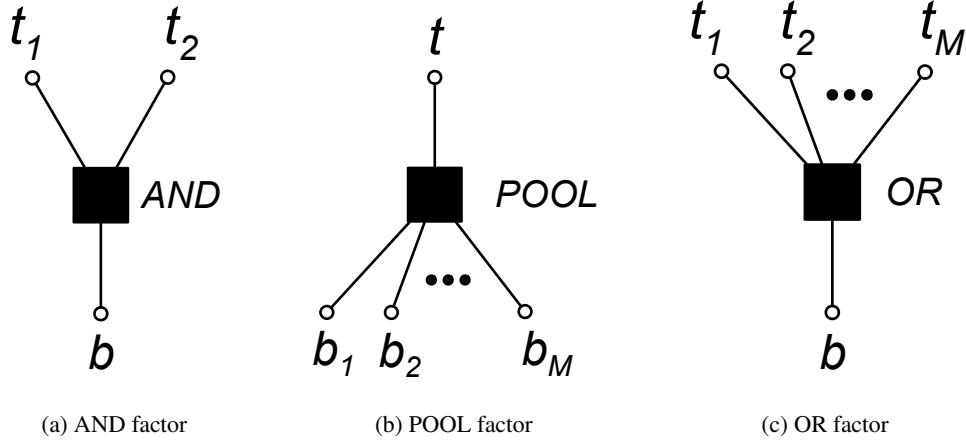

(a) AND factor       (b) POOL factor       (c) OR factor

Figure 11: Factors and variable labeling used in the message update equations.

### D.2 POOL FACTOR

Bottom-up message

$$\mathrm{m_{OUT}}(t) = \max(\mathrm{m_{IN}}(b_1), \ldots, \mathrm{m_{IN}}(b_M)) - \log M$$

Top-down messages

$$\mathrm{m_{OUT}}(b_m) = \min(\mathrm{m_{IN}}(t) - \log M, -\max_{j \neq m} \mathrm{m_{IN}}(b_j))$$

### D.3 OR FACTOR

Bottom-up messages

$$\mathrm{m_{OUT}}(t_m) = \min(\mathrm{m_{IN}}(b) + \sum_{j \neq m} \max(0, \mathrm{m_{IN}}(t_j)), \max(0, \mathrm{m_{IN}}(t_i)) - \mathrm{m_{IN}}(t_i)) \quad \text{with } i = \operatorname*{argmax}_{i \neq m} \mathrm{m_{IN}}(t_i)$$

Top-down message

$$\mathrm{m_{OUT}}(b) = \mathrm{m_{IN}}(t_i) + \sum_{j \neq i} \max(0, \mathrm{m_{IN}}(t_j)) \quad \text{with } i = \operatorname*{argmax}_{m} \mathrm{m_{IN}}(t_m)$$

## E    IMAGE CORRUPTION TYPE ILLUSTRATION

The different types of image corruption used in Section 4.4 are shown in the following Figure:

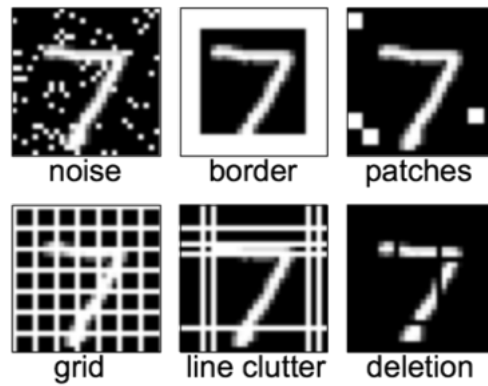

Figure 12: Different types of noise corruption used in Section 4.4.

