# Peer review of "Hierarchical compositional feature learning"

_ICLR 2017 — rejected_

[Official Review · AnonReviewer2 · rating 4 · confidence 4 · 10 Dec 2016]
**Interesting approach to compositional image modeling**

This paper presents a generative model for binary images.  Images are composed by placing a set of binary features at locations in the image.  These features are OR'd together to produce an image.  In a hierarchical variant, features/classes can have a set of possible templates, one of which can be active.  Variables are defined to control which template is present in each layer.  A joint probability distribution over both the feature appearance and instance/location variables is defined.

Overall, the goal of this work is interesting -- it would be satisfying if semantically meaningful features could be extracted, allowing compositionality in a generative model of images.  However, it isn't clear this would necessarily result from the proposed process.
Why would the learned features (building blocks) necessarily semantically meaningful?  In the motivating example of text, rather than discovering letters, features could correspond to many other sub-units (parts of letters), or other features lacking direct semantic meaning.

The current instantiation of the model is limited.  It models binary image patterns.  The experiments are done on synthetic data and MNIST digits.  The method recovers the structure and is effective at classification on synthetic data that are directly compositional.  On the MNIST data, the test errors are quite large, and worse than a CNN except when synthetic data corruption is added.  Further work to enhance the ability of the method to handle natural images or naturally occuring data variation would enhance the paper.

[Official Review · AnonReviewer3 · rating 5 · confidence 4 · 15 Dec 2016]
**This paper tackles a very interesting topic. However, it makes a false claim and a discussion/comparison to existing work is necessary. Experiments on real images will also strengthen the current submission**

This paper presents an approach to learn object representations by composing a set of templates which are leaned from binary images. 
In particular, a hierarchical model is learned by combining AND, OR and POOL operations. Learning is performed by using approximated inference with MAX-product BP follow by a heuristic to threshold activations to be binary. 

Learning hierarchical representations that are interpretable is a very interesting topic, and this paper brings some good intuitions in light of modern convolutional neural nets. 

I have however, some concerns about the paper:

1) the paper fails to cite and discuss relevant literature and claims to be the first one that is able to learn interpretable parts. 
I would like to see a discussion of the proposed approach compared to a variety of papers e.g.,:

- Compositional hierarchies of Sanja Fidler
- AND-OR graphs used by Leo Zhu and Alan Yuille to model objects
- AND-OR templates of Song-Chun Zhu's group at UCLA 

The claim that this paper is the first to discover such parts should be removed. 

2) The experimental evaluation is limited to very toy datasets. The papers I mentioned have been applied to real images (e.g., by using contours to binarize the images). 
I'll also like to see how good/bad the proposed approach is for classification in more well known benchmarks. 
A comparison to other generative models such as VAE, GANS, etc will also be useful.

3) I'll also like to see a discussion of the relation/differences/advantages of the proposed approach wrt to sum product networks and grammars.

Other comments:

- the paper claims that after learning inference is feed-forward, but since message passing is used, it should be a recurrent network. 

- the algorithm and tech discussion should be moved from the appendix to the main paper

- the introduction claims that compression is a prove for understanding. I disagree with this statement, and should be removed. 

- I'll also like to see a discussion relating the proposed approach to the Deep Rendering model. 

- It is not obvious how some of the constraints are satisfied during message passing. Also constraints are well known to be difficult to optimize with max product. How do you handle this?

- The learning and inference algorithms seems to be very heuristic (e.g., clipping to 1, heuristics on which messages are run). Could you analyze the choices you make?

- doing multiple steps of 5) 2) is not a single backward pass 

I'll reconsider my score in light of the answers

[Official Review · AnonReviewer1 · rating 5 · confidence 4 · 17 Dec 2016]
**My thoughts**

The paper discusses a method to learn interpretable hierarchical template representations from given data. The authors illustrate their approach on binary images.

The paper presents a novel technique for extracting interpretable hierarchical template representations based on a small set of standard operations. It is then shown how a combination of those standard operations translates into a task equivalent to a boolean matrix factorization. This insight is then used to formulate a message passing technique which was shown to produce accurate results for these types of problems.

Summary:
———
The paper presents an novel formulation for extracting hierarchical template representations that has not been discussed in that form. Unfortunately the experimental results are on smaller scale data and extension of the proposed algorithm to more natural images seems non-trivial to me.

Quality: I think some of the techniques could be described more carefully to better convey the intuition.
Clarity: Some of the derivations and intuitions could be explained in more detail.
Originality: The suggested idea is reasonable but limited to binary data at this point in time.
Significance: Since the experimental setup is somewhat limited according to my opinion, significance is hard to judge.

Details:
———
1. My main concern is related to the experimental evaluation. While the discussed approach is valuable, its application seems limited to binary images at this point in time. Can the authors comment?

2. There are existing techniques to extract representations of images which the authors may want to mention, e.g., work based on grammars.

[Final Decision · Program Chairs · 06 Feb 2017]
**ICLR committee final decision**

The reviewer's opinions were clear for this paper. Mainly it seems that the fact that this work focuses on binary image patterns limited the ability of reviewers to assess the significance of this work based on the instantiation of the model explored in this work. It was also noted that the writing could have been clearer when describing the intuitions for the approach and that the derivations could have been explained in more detail.